# SIDDESIGN: SEQUENCE-INFORMED DISTILLATION FOR TERTIARY STRUCTURE-BASED RNA DESIGN

## ABSTRACT

Tertiary structure-based RNA design, which aims to design nucleotide sequences that fold into a given 3D structure, is a fundamental challenge in synthetic biology and structure-guided design. Although recent work has advanced geometric encoders and architectural innovations, progress remains constrained by the scarcity and bias of resolved RNA tertiary structures. Considering that RNA folding and design are approximate inverse tasks, and that RNA foundation models employed in folding models capture rich RNA priors from large-scale sequence data, we are inspired to exploit these representation-level priors to enhance the RNA design task. Motivated by this perspective, we propose the **S**equence-**I**nformed **D**istillation framework for structure-based RNA **Design** (**SIDDesign**), which aligns structure-derived embeddings with sequence-level representations obtained from RNA foundation models. To bridge modality and contextual gaps between sequence and structure representations, we design a Similarity-aware Contextual Refinement (SimACR) module based on cross-attention. To further mitigate edge noise introduced during graph construction, we introduce a Stochastic Topology Regularization (STR) strategy during training that improves the robustness of message passing. Extensive experiments on benchmark datasets demonstrate the effectiveness of SIDDesign, with consistent and significant improvements over existing approaches.

## 1 INTRODUCTION

Ribonucleic acid (RNA) is a multifunctional molecule essential to numerous cellular activities, extending far beyond its classical role in genetic information transfer. Acting as a regulator, structural scaffold, and catalytic agent, RNA has become a versatile target for molecular engineering (Guo et al., 2010). Among emerging tasks, structure-to-sequence RNA design (Zuker, 2003; Hofacker et al., 1994; Yesselman & Das, 2015; Ken et al., 2023), which seeks to generate RNA sequences that adopt a predefined RNA secondary or tertiary structure, has attracted growing interest across synthetic biology, biotechnology, and therapeutic development. By exploiting certain structural constraints, the design paradigm enables the creation of spatially precise and functionally diverse RNA molecules, paving the way for applications ranging from programmable gene control to RNA-guided molecular assembly and targeted therapeutics (Warner et al., 2018; Hopkins & Groom, 2002).

Tertiary structure-based RNA design, which aims to generate sequences folding into a target 3D structure, is a critical but highly constrained problem. Recent studies (Tan et al., 2024; Huang et al., 2024; Joshi et al., 2025) have advanced structural modeling using geometric neural networks and diffusion-based generative frameworks, extracting features from atomic coordinates or ensembles of conformations. However, progress is fundamentally limited by the scarcity and bias of RNA tertiary structures. Unlike proteins, where large collections of sequence–structure pairs exist, publicly available RNA structures are rare and heavily skewed toward ribosomal complexes. As a result, models trained exclusively on such datasets are inherently constrained, since even sophisticated architectures struggle to generalize across the full diversity of RNA sequence–structure relationships. Although contrastive learning has been explored to enhance representation learning (Tan et al., 2024), these approaches still depend on the same restricted data, leaving the core scarcity challenge unresolved.

A complementary perspective arises when the problem is reframed through the lens of representation learning. Design and folding are not independent tasks but rather inverse mappings of the same

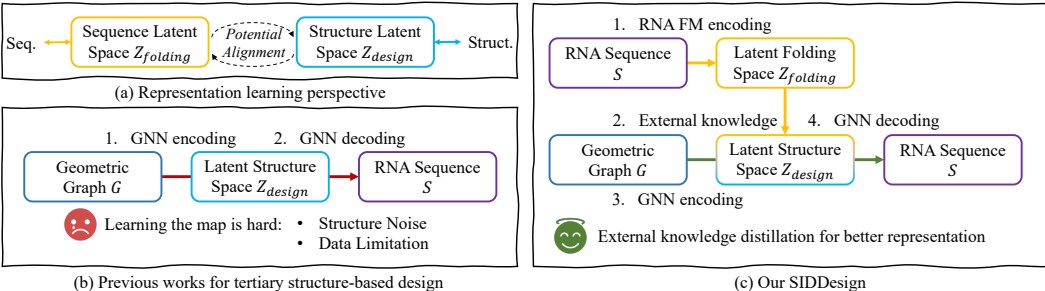

Figure 1: (a) Representation-learning view of folding and design. (b) Previous approaches focus solely on modeling and architecture, but the learning is challenging due to severe data limitations. (c) Our proposed SIDDesign leverages RNA foundation models to provide sequence-informed latent priors and distills them into structure-derived representations, yielding better performance.

underlying process. Folding models encode sequences into latent spaces and decode them into structures, often using SE(3)-equivariant mechanisms such as invariant attention. Design models, in contrast, attempt to traverse the reverse direction by mapping structures into latent spaces from which sequences can be generated. From this perspective, both tasks fundamentally depend on the geometry of their latent representations, as illustrated in Fig. 1(a). Most folding models today are built upon RNA foundation models trained on massive sequence corpora. Trained on large-scale sequence data, these models learn latent representations that encompass structural regularities as well as functional and evolutionary constraints. These sequence-informed representations provide an abundant external priors that can enrich the representation learning of design models. If the latent spaces of folding and design can be aligned, design models can inherit richer priors from RNA foundation models, thereby alleviating data scarcity and ultimately improving both the performance and robustness of sequence design.

Motivated by the above analysis, we propose to distill external priors from RNA foundation models into structure-derived representations, as illustrated in Fig. 1(c). This strategy offers a principled pathway to alleviate the data bottleneck in tertiary structure-based RNA design. Although aligning sequence and structure latent spaces appears natural, the inherent heterogeneity between modalities often leads to mismatched representations and unstable training. To address this issue, we introduce a Similarity-aware Contextual Refinement (SimACR) module that enriches sequence embeddings with structural context before alignment, ensuring that the distillation process preserves cross-modal semantics. The refined embeddings are then aligned with structural representations, thereby transferring large-scale sequence knowledge into the structure-side latent space and guiding representation learning in data-scarce regimes. Beyond latent alignment, structural modeling itself introduces additional challenges. Geometric graphs are generally constructed using fixed $k$-nearest-neighbor heuristics over atomic coordinates, which provide computational efficiency and maintain global receptive fields. However, the reliance on a fixed $k$ inevitably introduces noisy or spurious edges, and such artifacts can propagate through successive GNN layers, leading to overfitting and reduced generalization. To mitigate these effects, we design a Stochastic Topology Regularization (STR) strategy that randomly removes edges during training, serving as a form of structural data augmentation. This stochastic perturbation reduces the impact of graph noise, alleviates overfitting, and ultimately yields more robust structural representations.

In summary, our contributions are three-fold: (1) In this paper, we propose **S**equence-**I**nformed **D**istillation (SIDDesign) a framework that leverages abundant sequence-level priors from RNA foundation models to guide representation learning for tertiary structure-based RNA design. (2) In SIDDesign, we design a similarity-aware contextual refinement module that refines RNA-FM representations with structural context prior to alignment, thereby mitigating cross-modal mismatch. In addition, to address noisy edges introduced by $k$-nearest-neighbor graph construction, we introduce stochastic topology regularization, a training strategy that randomly perturbs graph connectivity to regularize message passing and improve robustness. (3) We conduct extensive experiments on benchmark datasets, showing that SIDDesign consistently outperforms competitive baselines, while ablation studies verify the effectiveness of each proposed component.

## 2 RELATED WORKS

**Tertiary Structure-based RNA Design.** RNA structure-to-sequence design has long been studied in computational biology due to its applications in synthetic biology, therapeutics, and biotechnology. Classical approaches relied on thermodynamic and energy-based optimization, including RNAfold (Lorenz et al., 2011), Mfold (Zuker, 2003), RNAInverse (Hofacker et al., 1994), and NU-PACK (Zadeh et al., 2011), which primarily targeted secondary structures under minimum free energy assumptions. Subsequent methods introduced stochastic or physically informed optimization (Yesselman & Das, 2015), but struggled with RNA's conformational flexibility, motivating the shift toward tertiary structure-based design (Ken et al., 2023). With the advent of deep learning, a variety of approaches have been proposed, ranging from reinforcement learning for secondary structures, such as LEARNA and Meta-LEARNA (Runge et al., 2018), to geometric networks and generative frameworks for tertiary structures, including RDesign (Tan et al., 2024), RiboDiffusion (Huang et al., 2024), gRNAde (Joshi et al., 2025), and RhoDesign (Wong et al., 2024).

**RNA Foundation Models.** Recent years have seen rapid progress in RNA foundation models through large-scale self-supervised pretraining on RNA sequences. Early efforts such as RNA-FM (Chen et al., 2022) and UNI-RNA (Wang et al., 2023) applied BERT-style masked language modeling on millions of non-coding RNAs, showing that latent evolutionary and structural signals can be captured without explicit labels. RiNALMo (Penić et al., 2024) further scaled model size to 650M parameters, both confirming that larger data and models yield stronger RNA representations. Beyond scaling, several works introduced structural or biological priors (Yin et al., 2024; Zhang et al., 2024b; Chu et al., 2024; Wang et al., 2024; Gong & Bu, 2024), such as base-pair aware masking in ERNIE-RNA (Yin et al., 2024) and motif-aware strategies in RNAErnie (Wang et al., 2024). More recent directions emphasize efficiency and cross-modality. For instance, ProtRNA (Zhang et al., 2024a) transferred protein LM priors, BiRNA-BERT (Tahmid et al., 2024) adopted adaptive tokenization for long transcripts, and Evo (Nguyen et al., 2024) scaled to a 7B-parameter genomic LM spanning DNA, RNA, and proteins.

**Cross-Modal Knowledge Distillation.** Several works have provided theoretical foundations for cross-modal knowledge distillation (KD). The modality focusing hypothesis (Xue et al., 2022) analyzed why cross-modal transfer may fail by identifying decisive features for effective distillation, while C2KD (Huo et al., 2024) introduced bidirectional distillation with selective transfer to mitigate modality mismatch. Earlier, Knowledge-as-Priors (Zhao et al., 2020) formulated a meta-learning approach to generalize superior modality priors to weaker modalities. Building on these foundations, research has explored practical applications. In 2D–3D perception, CMKD (Hong et al., 2022) transferred LiDAR knowledge into monocular detectors, X3KD (Klingner et al., 2023) extended this to multi-camera 3D detection with multi-level distillation, LabelDistill (Kim et al., 2024) employed label-guided supervision, and VeXKD (Ji et al., 2024) integrated sensor fusion with KD. For 2D/3D scene understanding, xMOD (Lahlali et al., 2025) distilled motion cues from 2D video into 3D discovery models via scene completion. In bio-sensing, Visual-to-EEG KD (Zhang et al., 2022) transferred expressive facial features into EEG-based models for emotion recognition. These advances highlight the versatility of KD across theory and applications.

## 3 METHODOLOGY

### 3.1 PRELIMINARIES

**Representation Learning Perspective on RNA Folding and Design.** Given an RNA represented as a geometric graph $G = (V, E)$ and sequence $S = (s_1, ..., s_n), s_i \in \{A, U, C, G\}$, tertiary structure-based design aims to recover a sequence $S'$ that is compatible with the given structure $G$. From the perspective of representation learning, this task amounts to learning a mapping between geometric and sequential modalities. In this view, structural encoding and sequence decoding are conceptually mediated by a latent representation that bridges the two domains.

$$G \xrightarrow{\text{encoding}} \mathbf{Z}_{\text{design}} \xrightarrow{\text{decoding}} S.$$

On the other hand, RNA folding follows the reverse process: sequences are first encoded through a pretrained RNA foundation model, then lifted into a structure-aware latent space via SE(3)-

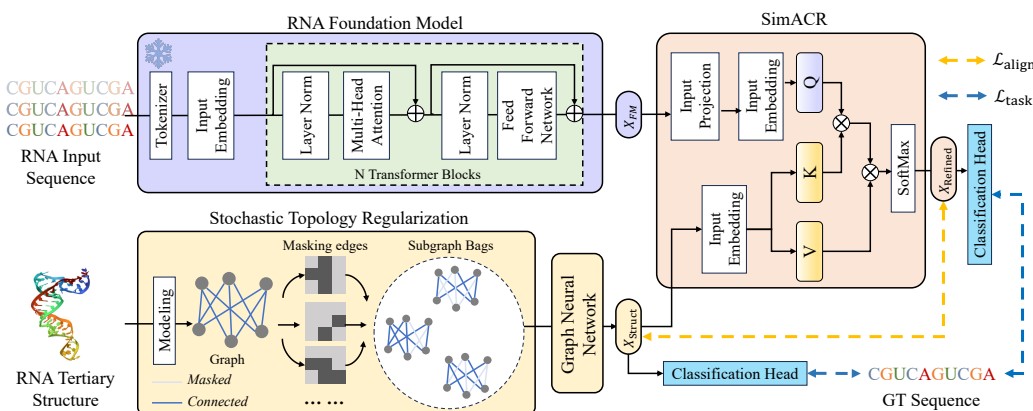

Figure 2: The overall pipeline of our SIDDesign.

equivariant geometry, and finally decoded into structures:

$$S \xrightarrow{\text{encoding}} \mathbf{Z}_{\text{folding},0} \xrightarrow{\Phi_{\text{SE(3)}}} \mathbf{Z}_{\text{folding},1} \xrightarrow{\text{decoding}} G.$$

Tertiary structure-based design is also known as RNA inverse folding. These two formulations highlight that folding and design are approximate inverses, connected through latent spaces that must capture sequence–structure compatibility. Effective design thus requires aligning structure-derived latent representations with sequence-informed latent before decoding. This need for a well-structured latent space motivates us to leverage external knowledge from RNA-FMs, using their rich sequence embeddings to guide and regularize representation learning in low-data regimes.

**RNA-FM Representation as External Priors.** To construct the sequence-informed latent space discussed above, we leverage large-scale pretrained RNA foundation models (RNA-FMs) as an external source of prior knowledge. Trained with masked token prediction objectives over massive RNA sequence corpora, these models have been shown to implicitly encode structural, functional, and evolutionary features within their internal representations. In our framework, RNA-FMs therefore provide a principled and abundant source of sequence-level information to guide structure-side representation learning.

Given the RNA sequence $S$, we extract contextual representations from the pretrained RNA foundation model, and then project it to the latent dimension compatible with the structure encoder:

$$\mathbf{Z}_{\text{seq}} = \mathcal{M}_{\text{RNAFM}}(S) \cdot W_{\text{proj}},$$

where $\mathcal{M}_{\text{RNAFM}}$ denotes the RNA foundation model and $W_{\text{proj}}$ is a linear projection. Since both sequence and structure representations are residue-aligned, $\mathbf{Z}_{\text{seq}}$ can serve as a one-to-one teacher for the structure encoder's representation $\mathbf{Z}_{\text{struct}}$. This sequence-informed embedding encodes biochemical and structural regularities that are difficult to learn from limited structure–sequence pairs alone, while also providing a strong latent geometry toward which structure-derived representations can be aligned.

## 3.2 SIDDESIGN

In this paper, we present Sequence-Informed Distillation (SIDDesign) for tertiary structure-based RNA design, a framework that integrates external sequence priors to enable robust latent space learning. SIDDesign is built on two key components: a similarity-aware contextual refinement module, which aligns structure-side representations with RNA-FM embeddings for cross-modal distillation, and a stochastic topology regularization strategy, which improves generalization by mitigating noise introduced during graph construction. The overall framework is illustrated in Fig. 2.

Given an RNA structure $G$, we first construct a geometric graph $G = (V, E)$ using the 3D coordinates following RDesign (Tan et al., 2024). After graph construction, the structure is encoded and decoded through multiple message passing layers, which update node representations by aggregating both geometric edge features and contextual information from neighboring residues. Finally, a

classification head predicts the nucleotide identity for each position, producing the complete RNA sequence in an end-to-end manner.

**Similarity-aware Contextual Refinement.** Since RNA-FM representations and structure-derived representations originate from distinct modalities with different input forms, a direct alignment is fundamentally limited. Furthermore, RNA-FM representations are generated through stacked attention layers, where each token integrates information from the entire sequence via global receptive fields. In contrast, structure-derived representations are processed by local message passing, where each node aggregates information only from its immediate neighbors based on 3D geometry. This discrepancy in contextual scope leads to another inherent semantic asymmetry: for any given token, the RNA-FM representation is context-rich but relatively abstract without structural grounding, whereas the GNN representation is grounded in local structure but context-limited. As a result, pointwise matching between $\mathbf{Z}_{\text{seq}}$ and $\mathbf{Z}_{\text{struct}}$ suffers from modality mismatch and unaligned contextual semantics.

To address this issue, we introduce a contextual refinement mechanism that recalibrates RNA-FM embeddings with structural context, ensuring semantic compatibility between refined sequence representations and structure-derived representations.

Formally, given RNA-FM representations $\mathbf{Z}_{\text{seq}} \in \mathbb{R}^{n \times d}$ and structure-side representations $\mathbf{Z}_{\text{struct}} \in \mathbb{R}^{n \times d}$, we compute a refined representation via cross-attention:

$$\mathbf{Z}_{\text{refined}} = \text{CrossAttn}(Q = \mathbf{Z}_{\text{seq}}, K = \mathbf{Z}_{\text{struct}}, V = \mathbf{Z}_{\text{struct}}),$$

where each token in $\mathbf{Z}_{\text{seq}}$ attends to the structure context to reweight its representation. We then leverage mean squared error to align the $\mathbf{Z}_{\text{refined}}$ and $\mathbf{Z}_{\text{struct}}$:

$$\mathcal{L}_{\text{align}} = ||\mathbf{Z}_{\text{refined}} - \mathbf{Z}_{\text{struct}}||_2^2,$$

which encourages the structure encoder to learn representations consistent with large-scale sequence priors. This mechanism not only regularizes the structure encoder but also transfers knowledge from large-scale sequence pretraining into the geometry domain, thereby improving latent representation learning for downstream RNA design.

**Stochastic Topology Regularization for Robust Message Passing.** While contextual refinement improves the semantic compatibility of the structure-side latent space, the encoder remains vulnerable to structural noise introduced during graph construction. In the modeling process, the graph is constructed using a fixed $k$-nearest-neighbor (kNN) strategy over 3D coordinates. This approach balances locality and efficiency while allowing certain long-range interactions to be captured. However, it may also introduce spurious edges that do not correspond to meaningful biochemical contacts, particularly when distant nodes are forcibly connected to satisfy the fixed degree. Although 3D distance provides a physically grounded approximation of interaction likelihood, RNA molecules are flexible and highly context-dependent: their folded tertiary structures are often shaped by macromolecular environments such as protein binding, which are not reflected in isolated structural data. As a result, the constructed graph can deviate from the true functional topology and exhibit noisy or unstable connections.

To address this, we introduce a stochastic topology regularization strategy during distillation training. At each training iteration, we sample a corrupted subgraph $\tilde{G} = (V, \tilde{E})$ by randomly dropping edges, and perform message passing over this subgraph $\tilde{G}$. This stochastic perturbation acts as a form of structural data augmentation, reducing sensitivity to noisy local topology and preventing overfitting. However, random edge removal alters node degree distributions and can destabilize feature propagation when a fixed normalization scale is used. To counter this, we employ degree-consistent normalization, in which aggregation weights are dynamically scaled according to the effective degree in the sampled subgraph:

$$\tilde{d}_i = |\tilde{E}(i)|,$$

$$h_i^{(l+1)} = \frac{1}{\tilde{d}_i} \sum_{j \in \tilde{E}(i)} \phi(h_i^{(l)}, h_j^{(l)}, e_{ij}),$$

where $\phi(\cdot)$ is the message passing function at layer $l$. This stochastic regularization, combined with degree-aware scaling, improves the robustness and generalization of the structure encoder under noisy or imperfect graph topologies, complementing the semantic alignment achieved through contextual distillation.

### 3.3 Training Objective

During training, both the structure-derived representation $\mathbf{Z}_{\text{struct}}$ and the refined RNA-FM representation $\mathbf{Z}_{\text{refined}}$ are passed through a shared linear readout head $f_{\text{readout}}$ to produce per-token logits over the nucleotide vocabulary. The task objective is defined as a token-wise cross-entropy loss:

$$\mathcal{L}_{\text{task}} = \frac{1}{n} \sum_{i=1}^{n} [\text{CE}(f_{\text{readout}}(\mathbf{z}_{\text{struct},i}), s_i) + \alpha \cdot \text{CE}(f_{\text{readout}}(\mathbf{z}_{\text{seq},i}), s_i)],$$

where $\alpha$ balances the supervision strength between the structure encoder and the RNA-FM branch. To improve robustness against geometric perturbations and reduce overfitting, isotropic Gaussian noise is injected into the input 3D coordinates prior to graph construction, following previous practice. Specifically, the perturbed coordinates are given by

$$\tilde{\mathbf{X}}_{\text{coord}} = \mathbf{X}_{\text{coord}} + \epsilon \cdot \mathcal{N}(0, \mathbf{I}),$$

where we can adjust the noise magnitude by $\epsilon$. During training, $\epsilon = 0.1$, which approximately yields Root-Mean-Square Deviation (RMSD) perturbations within $0.2 - 0.4\text{Å}$, consistent with the fluctuation range typically observed in experimental and simulation-derived RNA structures.

The final training objective combines the task loss and alignment loss as

$$\mathcal{L}_{\text{total}} = \mathcal{L}_{\text{task}} + \beta \cdot \mathcal{L}_{\text{align}},$$

where $\beta$ controls the relative strength of the alignment loss. This unified strategy enables the model to decode sequences from structural inputs while remaining semantically aligned with large-scale sequence priors and resilient to coordinate-level perturbations. It is worth noting that during inference, both the RNA-FM branch and the similarity-aware contextual refinement module are removed, leaving only the GNN encoder–decoder pathway. As a result, SIDDesign introduces no additional computational overhead at inference time.

## 4 Experiment

### 4.1 Datasets and Implementation

We evaluate our SIDDesign on the benchmark dataset introduced by RDesign (Tan et al., 2024). Results are reported for three length regimes (short, medium, and long) and for the entire benchmark. We compare against sequence-only recurrent baselines (SeqRNN and SeqLSTM with hidden sizes 128 and 256), structure-based feed-forward and message-passing models, StructMLP, StructGNN, GraphTrans (Ingraham et al., 2019), PiFold Gao et al. (2023), gRNAde[1] (Joshi et al., 2025), and RDesign (Tan et al., 2024), which augments structural representations through perturbation-based contrastive learning.

We implement our SIDDesign using PyTorch and adopt exactly the same geometric graph construction and backbone architecture MPNN (Dauparas et al., 2022) as RDesign to ensure a fair comparison. SIDDesign adds two ingredients: the similarity-aware contextual refinement for cross-modal distillation and stochastic topology regularization during training. We report mean±standard deviation over repeated runs with different random seeds. All experiments are conducted on a single NVIDIA A100 GPU. Model training is performed using the AdamW optimizer with an initial learning rate of $1 \times 10^{-3}$ and a batch size of 64. For hyper-parameters, the teacher-branch task loss weight is set to $\alpha = 1.0$ and the distillation weight is set to $\beta = 2.0$. For stochastic topology regularization, edges are randomly dropped with a probability of 30%. At inference, both the teacher-branch and the contextual refinement module are removed, leaving only the GNN pathway, thus, SIDDesign introduces no additional computational overhead.

### 4.2 Main Results

**Results on benchmark dataset.** Tab. 1 summarizes the recovery rate of different methods across RNA sequences of varying lengths. Our SIDDesign achieves 45.27%±1.56 on short sequences,

---

[1]Since gRNAde originally used a different dataset, we re-implement its results. For a fair comparison, we replace the coordinates used for graph construction in gRNAde (P, C4', N1/N9) with those adopted in RDesign (P, O5', C5', C4', C3', O3'). While results for the others are directly taken from the original paper of RDesign.

Table 1: The recovery rate on the benchmark dataset. The best results are highlighted in bold.

| Method | Short | Medium | Long | All |
|---|---|---|---|---|
| SeqRNN (h=128) | 26.52±1.07 | 24.86±0.82 | 27.31±0.41 | 26.23±0.87 |
| SeqRNN (h=256) | 27.61±1.85 | 27.16±0.63 | 28.71±0.14 | 28.24±0.46 |
| SeqLSTM (h=128) | 23.48±1.07 | 26.32±0.05 | 26.78±1.12 | 24.70±0.64 |
| SeqLSTM (h=256) | 25.00±0.00 | 26.89±0.35 | 28.55±0.13 | 26.93±0.93 |
| StructMLP | 25.72±0.51 | 25.03±1.39 | 25.38±1.69 | 25.35±0.25 |
| StructGNN | 27.55±0.94 | 28.78±0.87 | 28.23±1.95 | 28.23±0.71 |
| GraphTrans | 26.15±0.93 | 23.78±1.11 | 23.80±1.69 | 24.73±0.93 |
| PiFold | 24.81±2.01 | 25.90±1.56 | 23.55±1.43 | 24.48±1.13 |
| RDesign | 37.22±1.14 | 44.89±1.67 | 43.06±0.08 | 41.53±0.38 |
| gRNAde | 42.40±1.41 | **45.10±0.92** | 43.42±1.62 | 43.26±0.71 |
| Ours | **45.27±1.56** | 44.49±0.58 | **44.28±1.27** | **44.85±0.90** |

Table 2: The Macro-F1 on the benchmark dataset. The score is multiplied by 100 for aesthetics.

| Method | Short | Medium | Long | All |
|---|---|---|---|---|
| SeqRNN (h=128) | 17.22±1.69 | 17.20±1.91 | 8.44±2.70 | 17.74±1.59 |
| SeqRNN (h=256) | 12.54±2.94 | 13.64±5.24 | 8.85±2.41 | 13.64±2.69 |
| SeqLSTM (h=128) | 9.89±0.57 | 10.44±1.42 | 10.71±2.53 | 10.28±0.61 |
| SeqLSTM (h=256) | 9.26±1.16 | 9.48±0.74 | 7.14±0.00 | 10.93±0.15 |
| StructMLP | 17.46±2.39 | 18.57±3.45 | 17.53±8.43 | 18.88±2.50 |
| StructGNN | 24.01±3.62 | 22.15±4.67 | 26.05±4.63 | 24.87±1.65 |
| GraphTrans | 16.34±2.67 | 16.39±4.74 | 18.67±7.16 | 17.18±3.81 |
| PiFold | 17.48±2.24 | 18.10±6.76 | 14.06±3.53 | 17.45±1.33 |
| RDesign | 38.25±3.06 | 40.41±1.27 | 41.48±0.91 | 40.89±0.49 |
| gRNAde | 39.22±0.59 | **45.22±1.23** | 42.21±1.30 | 43.09±0.39 |
| Ours | **41.60±0.37** | 44.24±0.52 | **42.74±0.66** | **43.30±0.04** |

44.49%±0.58 on medium sequences, and 44.28%±1.27 on long sequences, yielding an overall recovery rate of 44.85%±0.90, the best among all methods. Compared with RDesign (Tan et al., 2024), which adopts the same architecture but relies on contrastive learning, SIDDesign consistently delivers higher or similar recovery rate. On short sequences, SIDDesign achieves a recovery rate improvement of 8.05% over RDesign, while on long sequences it achieves a gain of 1.22%. This observation supports our analysis of graph noise, and the larger gain on short RNAs may stem from the limited number of nucleotides, which makes kNN graphs more likely to include spurious edges. With stochastic topology regularization, the model can better handle such edge noise, which in turn contributes to the improvement. On medium sequences, although RDesign and gRNAde performs competitively with 44.89% and 45.10%, the margin over our performance is small and comes with higher variance. Overall, across all sequence length ranges, SIDDesign maintains more balanced performance, achieving an overall improvement of +3.32% compared with RDesign and 1.59% compared with gRNAde. This improvement further underscores the advantage of leveraging RNA-FM priors to guide structure-side representation learning.

Tab. 2 reports the Macro-F1 results on the benchmark dataset. Our method achieves 41.60%±0.37, 44.24%±0.52, and 42.74%±0.66 on short, medium, and long sequences, respectively, leading to an overall score of 43.30%±0.04. Similar to the recovery rate results, SIDDesign consistently outperforms RDesign across all length ranges. Taken together, the improvements in both recovery rate and Macro-F1 confirm the benefit of distilling sequence-informed priors, underscoring the advantage of leveraging RNA-FM representations to guide structure-side representation learning. Combined with stochastic topology regularization, our approach achieves more reliable and balanced performance across different sequence lengths.

**Results for structural fidelity.** To assess the structural fidelity of designed sequences, we evaluate the structures obtained by folding them with Protenix. Fig. 3 presents both the overall RMSD

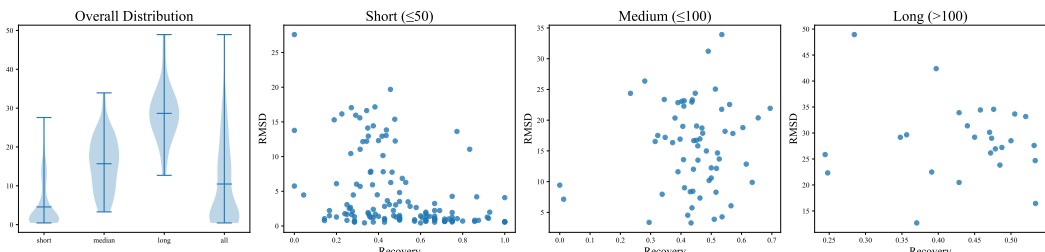

Figure 3: Structure fidelity of designed sequences, the designed sequences is folded by Protenix.

Table 3: The overall recovery rate and Macro-F1 scores on the Rfam and RNA-Puzzles datasets.

| Method | Recovery Rate | | Macro F1 | |
|---|---|---|---|---|
| | Rfam | RNA-Puzzles | Rfam | RNA-Puzzles |
| StructMLP | 24.40±1.63 | 24.22±1.28 | 16.79±4.01 | 16.40±3.28 |
| StructGNN | 27.64±3.31 | 27.96±3.08 | 23.84±3.45 | 22.51±4.15 |
| GraphTrans | 23.81±2.57 | 22.21±2.98 | 17.32±5.28 | 17.04±5.36 |
| PiFold | 22.55±4.13 | 23.78±6.52 | 16.08±2.34 | 16.32±4.62 |
| MCTS-RNA | 31.74±0.07 | 32.06±1.87 | 23.82±4.40 | 24.12±3.47 |
| LEARNA | 31.92±2.37 | 30.94±1.45 | 22.15±1.77 | 22.75±1.17 |
| aRNAque | 30.01±3.26 | 31.07±2.74 | 22.84±1.70 | 23.03±1.65 |
| eM2dRNAs | 33.34±1.02 | 37.10±3.24 | 24.80±3.88 | 26.91±2.32 |
| RDesign | 56.12±1.03 | 50.12±1.07 | 53.27±1.28 | 49.24±1.07 |
| gRNAde | 58.31±0.69 | 50.26±0.50 | 56.27±0.75 | 49.74±0.75 |
| Ours | **66.05±0.38** | **55.20±1.39** | **63.83±0.29** | **54.98±0.81** |

distribution and scatter plots across different length regimes. Across the full benchmark, the folded structures exhibit a mean RMSD of 10.46, indicating that the generated sequences generally preserve the global topology of the target backbones. When stratified by sequence length, short RNAs achieve substantially lower deviation (4.56 on average), while medium and long RNAs naturally present higher RMSD values (15.68 and 28.65, respectively), reflecting the increased flexibility for longer molecules. The corresponding TM-score further supports these findings, with an average score of 0.42, suggesting moderate structural similarity and consistent structural alignment.

**Results on generalization datasets.** To further examine generalization, we evaluate SIDDesign on two external benchmarks Rfam (Nawrocki et al., 2015) and RNA-Puzzles (Miao et al., 2020) which are used in RDesign. As reported in Tab. 3, our method achieves clear improvements over RDesign across both datasets. On the Rfam benchmark, SIDDesign achieves a recovery rate of 66.05% and a Macro-F1 of 63.83%, yielding improvements of 9.93% and 10.56%, respectively. Similarly, on RNA-Puzzles, our SIDDesign achieves improvements of 5.08% and 5.74%. Although both benchmarks contain relatively limited test samples, which may lead to larger standard deviations, the overall margins remain substantial, confirming the robustness of our approach. These consistent improvements demonstrate that SIDDesign not only performs well under in-domain evaluation but also generalizes effectively to challenging unseen RNA structures.

## 4.3 ABLATION STUDIES

**Analysis of the Effectiveness of Each Component.** We conduct ablation studies to verify the contribution of each component in SIDDesign, the overall results are shown in Tab. 4. Replacing the contrastive learning with a naive knowledge distillation improves the overall recovery rate from 41.53% to 42.06%, indicating the effectiveness of introducing sequence-informed priors. However, the performance over length regimes are uneven: while the medium-length sequences benefit, the short and long categories remain less performed. This observation corroborates our previous analysis that direct alignment suffers not only from modality discrepancy but also from mismatched receptive fields, leading to inconsistent performance across sequence lengths. Incorporating the

Table 4: Effectiveness of proposed modules on benchmark dataset. *pert.* denotes coordinate perturbation, and *TTA* represents test-time augmentation.

| Settings | SimACR | STR | Short | Median | Long | All |
|---|---|---|---|---|---|---|
| KD (w/ pert.) | | | 41.22±0.74 | 44.22±0.94 | 43.68±0.45 | 42.34±0.74 |
| KD (w/o pert.) | | | 40.30±0.71 | 45.26±1.74 | 43.60±0.40 | 42.06±0.10 |
| KD (w/ pert.) | ✔ | | 42.06±0.92 | 44.39±0.50 | 43.87±0.62 | 42.92±0.36 |
| KD (w/o pert.) | ✔ | | 42.27±1.89 | 44.47±1.60 | 42.79±1.17 | 42.92±0.60 |
| Ours (idx=2) | ✔ | ✔ | 44.85±0.15 | 44.33±0.53 | 44.85±0.27 | 44.72±0.22 |
| Ours (idx=3) | ✔ | ✔ | 45.02±1.04 | 44.16±0.70 | 44.28±0.72 | 44.69±0.62 |
| Ours (idx=5) | ✔ | ✔ | 43.73±0.48 | 43.64±0.52 | 44.12±1.12 | 43.75±0.28 |
| Ours | ✔ | ✔ | 45.27±1.56 | 44.49±0.58 | 44.28±1.27 | 44.85±0.90 |
| +TTA (ens=3) | ✔ | ✔ | 44.50±0.87 | 42.86±0.56 | 42.19±0.51 | 44.37±0.49 |
| +TTA (ens=5) | ✔ | ✔ | 46.51±0.84 | 43.13±0.45 | 42.90±0.26 | 44.82±0.19 |
| +TTA (ens=10) | ✔ | ✔ | 46.63±0.47 | 43.42±0.42 | 43.48±0.62 | 45.59±0.34 |

similarity-aware contextual refinement module (SimACR) addresses this issue. Although its average performance on certain ranges is slightly lower than naive KD, the results across short, medium, and long sequences are more balanced, confirming that context-aware recalibration reduces cross-modal mismatch and yields more consistent representation learning. Finally, adding stochastic topology regularization (STR) provides further improvements. With all proposed components, our SIDDesign achieves the best results across all length regimes, reaching 45.27%, 44.49%, and 44.28% on short, medium, and long sequences, respectively, and an overall recovery rate of 44.85%. This shows that both sequence-informed distillation and structural regularization are effective, enhancing not only performance but also robustness in tertiary structure-based RNA design.

**Effect of Distillation Layer Position.** We also examine how the choice of distillation layer within the MPNN network influences the final performance. As shown in Tab. 4, using the intermediate representations (idx=2/3) yields very similar performance, with negligible differences across all sequence-length ranges. In contrast, applying distillation to second-to-last layer (idx=5) noticeably reduces performance, especially on short sequences and in the overall average. This slightly worse performance may due to the classification loss applied to the teacher branch, this may lead to conflicting supervision signals. In contrast, when the distillation loss is imposed on a middle layer, several subsequent layers can further process and reconcile this supervision, similar to auxiliary deep supervision in vision models, so no strong conflict is observed.

**STR as Test-Time Augmentation.** We further evaluate the use of stochastic topology regularization as a test-time augmentation (TTA) strategy by applying random edge dropout during inference and ensembling predictions over 3, 5, and 10 stochastic forward passes. As shown in Tab. 4, TTA yields clear gains in the short sequence regime, where recovery improves from 45.27% to 46.63% with an ensemble of 10. This observation aligns well with our original motivation for STR, when constructing graphs with a fixed $k = 30$, short RNAs contain very few residues, leading the kNN procedure to produce an almost fully connected graph, which inevitably introduces a large number of spurious edges. Applying STR at inference effectively mitigates these noisy connections, making the model more robust to such noise and thus improving performance. In contrast, for medium and long sequences, the effect of TTA is nuanced. Since longer RNAs have richer geometric neighborhoods, kNN may already omit some meaningful long-range interactions. Introducing additional edge dropout during inference may therefore remove valid connections, resulting in slight performance decreases in these regimes.

**Impact of Distillation Weight.** In this part, we study the influence of the distillation weight $beta$, the results are shown in Fig. 4. For recovery rate, performance improves steadily as $beta$ increases, and the gain saturates around $\beta = 2.0$. For instance, the overall recovery rises from 44.11% at $beta = 0.5$ to 44.84% at $\beta = 2.0$, with a recovery rate of 45.28% on short sequences and 44.49% on median sequences. Macro-F1 exhibits a similar trend, the overall score peaks near $\beta = 2.0$. These results suggest that assigning sufficient weight to the distillation loss is essential for transferring RNA-FM priors, and we adopt $\beta = 2.0$ as it offers the best balance between accuracy and stability.

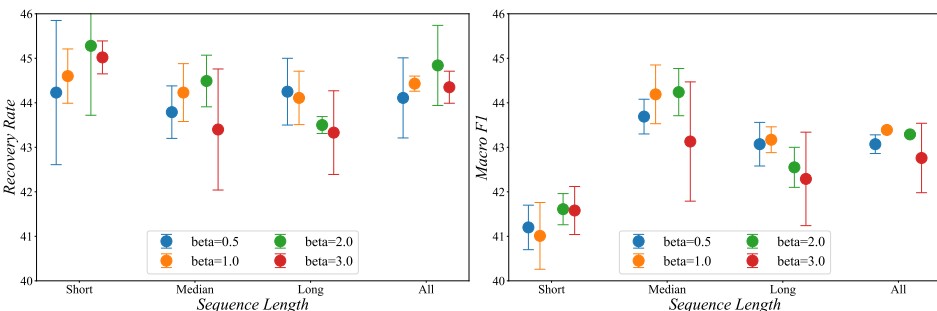

Figure 4: Ablation study on the effect of different distillation weight $\beta$.

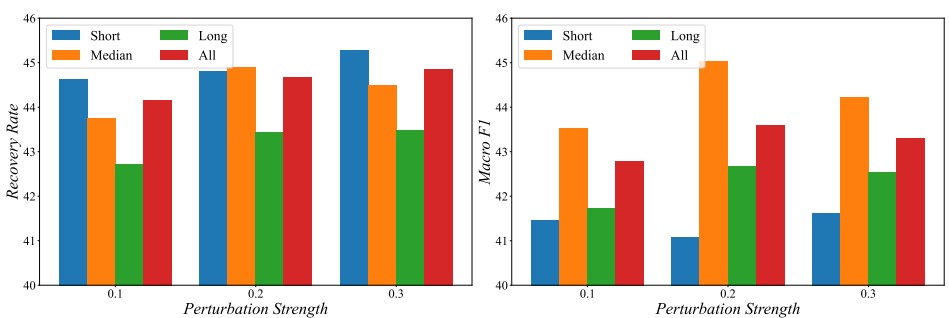

Figure 5: Ablation study on the effect of stochastic topology regularization.

**Impact of Strength of Stochastic Topology Regularization.** We further conduct several experiments to verify the impact of the edge-drop probability in stochastic topology regularization. As shown in Fig. 5, increasing the dropping probability generally improves performance across both recovery rate and Macro-F1. For recovery rate, the overall score grows from 44.17% at $p = 0.1$ to 44.85% at $p = 0.3$. Short sequences benefit the most, with an absolute gain of about +0.6% from weak to strong perturbation. Macro-F1 exhibits a similar trend, where $p = 0.2$ achieves the highest average score, particularly on medium-length sequences, but also shows larger variance, e.g., a standard deviation of 2.48. Overall, the results indicate that stronger perturbations enhance robustness, and we adopt $p = 0.3$ as it provides more stable performance across all length ranges.

## 5 CONCLUSION

In this paper, we introduced SIDDesign, a sequence-informed distillation framework for tertiary structure-based RNA design. Motivated by the representation learning perspective that folding and design are approximate inverses, SIDDesign leverages RNA-FMs to provide sequence-level priors and distills them into structure-derived representations via similarity-aware contextual refinement. To further enhance robustness, stochastic topology regularization is applied to alleviate edge noise introduced during graph construction. Extensive experiments verify the effectiveness of SIDDesign, showing clear improvements over previous methods. On benchmark datasets, it achieves better recovery rates and Macro-F1 scores with more balanced performance across different sequence lengths. Overall, SIDDesign provides a principled framework for leveraging sequence-informed priors to guide structure-based RNA design, leading to improvements in both performance and robustness. In future work, we plan to extend our framework beyond isolated RNA molecules to model RNA complexes. On the design side, we aim to incorporate diverse biochemical and structural constraints, such as affinity and secondary structure, into the generation process. By jointly considering these factors, our framework could be adapted to more realistic biological settings and unlock broader applications in RNA therapeutics, synthetic biology, and molecular engineering.

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

## ETHICS STATEMENT

This work develops a computational framework for tertiary structure-based RNA design and is conducted entirely on publicly available datasets, without involving human, clinical, or otherwise sensitive data. While the methods may contribute to advances in synthetic biology and therapeutics, any downstream applications must adhere to established biosafety standards and ethical guidelines. We are committed to transparency, fairness, and responsible stewardship in line with ICLR's Code of Ethics.

## REPRODUCIBILITY STATEMENT

Our experiments are based on the open-source implementations of RDesign and gRNAde, with training settings and hyper-parameters described or analyzed in the experimental section. All experiments were conducted with the same random seed sequence and repeated three times, reporting both mean and variance to ensure reproducibility.

## USE OF LARGE LANGUAGE MODELS

In preparing this manuscript, large language models (LLMs) are used solely for polishing and refining the writing, and not for generating or proposing any methodological content.

## A  QUALITATIVE ANALYSIS OF FEATURE ALIGNMENT WITH KD

To better understand the latent of representation learning and analyze how knowledge distillation reshapes the latent space, we visualize three types of representations:

- the original teacher embeddings from the RNA foundation model
- the teacher embeddings after applying SimACR refinement
- the student (MPNN) representations

Since the original RNA-FM representations differ in channel dimensionality, we first apply PCA to reduce each representation to a 64-dimensional space for alignment. The reduced features are then projected into two dimensions using t-SNE for visualization, allowing us to directly compare the geometric structure of the latent spaces before and after refinement.

As shown in Fig. 6, we first apply global average pooling to obtain a sample representation per RNA. The left plot shows the t-SNE projection of the student representations and the original teacher embeddings (without SimACR). Despite after KD training, a substantial gap remains between the two

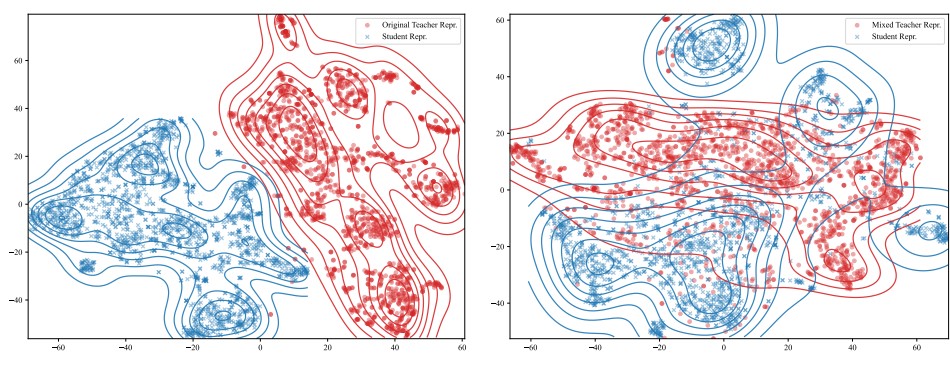

Figure 6: Sample level t-SNE of teacher and student representations before and after SimACR.

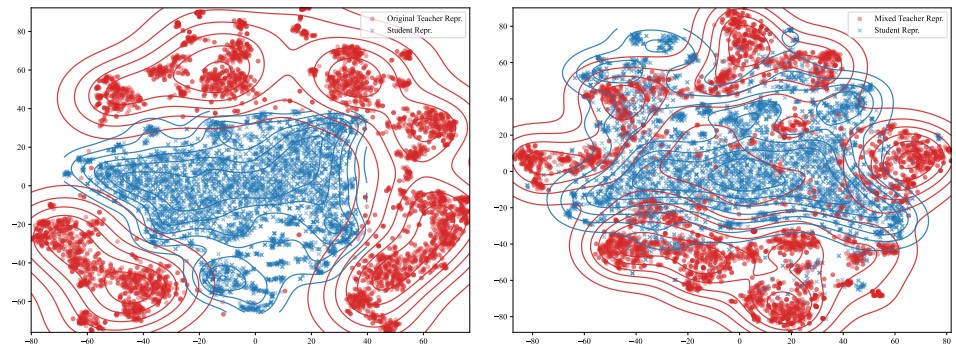

Figure 7: Nucleotide level t-SNE of teacher and student representations before and after SimACR.

representations. This discrepancy may arise from multiple factors, including the differing pretraining objectives of RNA foundation models, the inherent modality mismatch between geometry and sequence, and so on. In contrast, after applying the proposed SimACR module (right plot), the refined teacher embeddings exhibit significantly better overlap with the student representations.

We further visualize the t-SNE at the nucleotide level, where each token corresponds to a nucleotide-specific representation. As shown in Fig. 7, nucleotide-level distributions exhibit patterns distinct from the sample-level visualization. In particular, the teacher embeddings from the RNA foundation model display clear clustering behavior, reflecting the strong contextual priors learned during large-scale sequence pretraining. In the left plot, the gap between teacher and student representations remains substantial, with almost no overlap in the latent space. This indicates that the student encoder struggles to match the fine-grained, token-level semantics intrinsic to the RNA-FM representations under naive distillation. After performing SimACR, the refined teacher embeddings show noticeably improved overlap with the student representations. Although some discrepancy still remains, the two distributions become significantly closer while the intrinsic cluster structure of the teacher representations is preserved, demonstrating that SimACR reduces cross-modal mismatch without representational collapse.

## B  RESULTS UNDER SEQUENCE SIMILARITY SPLIT

To further evaluate the generalization ability of our model, we construct additional train–test split based on sequence similarity. We apply CD-HIT-EST with a similarity threshold of 0.8 to cluster and deduplicate sequences. After removing redundant sequences, the dataset contains 1697 unique sequences, which are grouped into 651 clusters. To avoid bias caused by uneven length distributions, we perform cluster-level grouping according to length distribution, ensuring that the final train/test split preserves a balanced length distribution. This results in 1351 training sequences and 346 test sequences. The corresponding sequence-length distribution is shown in the boxplot in Fig. 8.

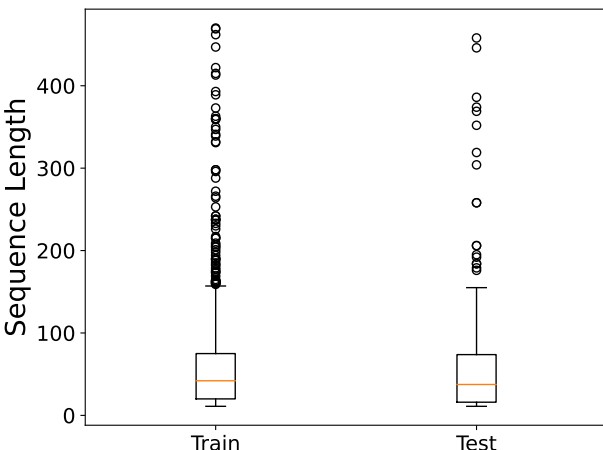

Figure 8: The sequence length distribution of train and test set under sequence similarity split.

Table 5: The recovery rate on the benchmark dataset under sequence similarty split. The best results are highlighted in bold. *: *For a fair comparison, we report results for RhoDesign (NAR), a non–autoregressive variant of RhoDesign. Specifically, instead of the original next-token–prediction (autoregressive) decoder, we replace it with an end-to-end NAR decoder so that the modeling setup is consistent with other methods. Moreover, we find that the AR decoder in the original RhoDesign is difficult to train under this data scale, often failing to converge within a reasonable number of iterations and likely requiring substantially longer training schedules. Furthermore, to ensure parameter-level fairness, we reduce the Transformer dimensionality in RhoDesign to 128 (one quarter of its original dim), resulting in a total parameter count of 1.3M~1.5M, comparable to GVP, MPNN, and other baselines included in this evaluation.*

| Method | Short | Medium | Long | All |
|---|---|---|---|---|
| MPNN Baseline | 42.12±0.75 | 49.06±0.59 | 48.75±0.07 | 44.95±0.35 |
| gRNAde | 40.98±0.75 | 48.08±0.13 | 48.72±0.49 | 44.01±0.44 |
| RhoDesign* (NAR) | 41.72±0.76 | 48.16±0.37 | 48.62±0.58 | 44.44±0.38 |
| Ours | **44.46±0.12** | **50.73±0.32** | **50.84±0.25** | **47.07±0.05** |

Tab. 5 reports the recovery performance under the sequence-similarity–based split constructed with CD-HIT-EST. Despite the different data partitioning protocol, our method continues to deliver the best performance across all length regimes. To be specific, SIDDesign achieves recovery rates of 44.46%, 50.73%, and 50.84% on short, medium, and long RNAs, respectively, outperforming MPNN Baseline, gRNAde, and RhoDesign (NAR) by clear margins. Overall, our approach reaches 47.07%, again the highest among all compared methods. These results demonstrate that our SID-Design remains robust under different generalization settings and continues to outperform existing other models.

