# OpenReview forum: "SIDDesign: Sequence-Informed Distillation for Tertiary Structure-Based RNA Design"
_ICLR.cc/2026/Conference — Submitted to ICLR 2026_

### Official Review · Reviewer_h4Nv · 2025-10-25

**Soundness:** 3
**Presentation:** 3
**Contribution:** 2
**Rating:** 4
**Confidence:** 4

**Summary:**

The paper proposes SIDDesign, a framework that transfers large-scale sequence priors from pretrained RNA foundation models into the structure-side latent space by aligning structure encoder embeddings with token-level teacher representations while tackling two core failure modes: a modality/context mismatch and spurious edges in kNN graphs. To narrow the modality gap, the Similarity-aware Contextual Refinement (SimACR) first injects structure context into sequence embeddings before alignment; to improve robustness to noisy topology, Stochastic Topology Regularization (STR) randomly perturbs graph connectivity during training with degree-consistent normalization. Implementation follows an existing RDesign-style geometric graph pipeline to ensure comparability, and at inference time, the teacher and refinement modules are removed, so there is no extra runtime cost. Experiments show consistent gains in recovery rate and Macro-F1 with more balanced performance across short, medium, and long sequences, particularly strong improvements on short sequences where graph noise is more pronounced, and the method generalizes well on external benchmarks such as Rfam and RNA-Puzzles.

**Strengths:**

- Rather than naively aligning embeddings, the method first refines sequence representations with structure context to correct the global-vs-local contextual asymmetry before alignment, which is a principled fix to the modality mismatch.

- The work explicitly tackles spurious edges from fixed-k kNN graphs using stochastic edge dropping and degree-aware normalization, yielding topology-level data augmentation and improved generalization.

- The authors follow the RDesign pipeline for geometric graph construction and message passing, isolating the gains to the proposed distillation and regularization modules.
- Gains appear in both recovery rate and Macro-F1 across length ranges, with especially strong improvements on short sequences where noisy edges are more likely.

**Weaknesses:**

- I have a very general question about those cross-modality biomolecule sequence design papers: The method leans heavily on RNA foundation model as the teacher for representation alignment, which risks importing that model’s inductive biases, such as dataset composition and pretraining objectives, into the structure-side space. The paper mentions that the sequence embedding is not aligned with the structural embedding in the beginning. Could the authors further elaborate on that in detail? For example, why/how are those embeddings aligned, and how could you say that it is not aligned? I think it is a very important question in the field. If you adopt the same MPNN scheme as RDesign or similar approaches, the Z_struct should have the same order as the Z_seq.


- Evaluation is dominated by token-level correctness rather than functional plausibility. Recovery rate and Macro-F1 are useful diagnostics, but the problem of inverse folding ultimately depends on whether generated sequences fold back to the target structure and sit in favorable energy ranges while respecting secondary-structure pairings and chemical constraints. Optimizing for per-position classification may overfit to local statistics and under-represent global folding consistency. The study would be more convincing if it included fold-back checks with at least some secondary-structure predictors, and if it reported how often designed sequences reach the correct topologies or satisfy known structural constraints.


- Diversity-accuracy trade-off is not quantified. In structure-based design there are typically many valid sequences per target; an approach that increases accuracy but collapses diversity can be ill-suited for downstream screening or optimization (like a sequence full of G/Cs). Distillation from a strong teacher can inadvertently narrow the output distribution, particularly if the alignment is tight and decoding is deterministic. The paper would benefit from sampling-based analyses (temperature, top-k/top-p), diversity metrics (e.g., unique sequence ratio, pairwise Hamming distances).

- The method’s success depends on where and how the refinement is inserted relative to the structure encoder and predictor head, but the paper provides limited analysis on numbers of layers or attention heads. Different placements can change the balance between local geometric fidelity and global sequence context, affect training stability, and carry different compute footprints. A careful ablation would help guide re-use in other architectures and clarify whether most gains come from a single well-placed refinement block or from stacking.

- During training, the model benefits from a teacher branch and from stochastic topology perturbations; at inference, both are absent. While such regularization patterns are common, the paper does not quantify how sensitive the final predictor is to the strength of these training-only signals, nor whether certain settings over-regularize and harm clean-graph inference. Reporting curves that vary teacher alignment strength and STR rates, together with inference-time robustness tests, would clarify stability and reveal whether the method is robust to mismatches between training and deployment conditions. As it might be time-consuming, I think the authors could only explain that part in more detail, the experiments could be added later.

**Questions:**

Please see the weaknesses part, I have included my questions in it.

---

> ### Author Response · Authors · 2025-11-21
>
> Thank you for your constructive comments. Here we give the point-to-point reply to the comments.
>
> ### 1. On representation alignment and embedding discrepancy
>
> Thank you for raising this important conceptual question.
> We added a detailed visualization analysis in Appendix A. Using t-SNE, we show RNA-FM embeddings before/after SimACR, together with the student GNN representations. The plots consistently demonstrate that the native RNA-FM space and structural embeddings remain noticeably separated, even after KD.
> This gap may stem from various reasons (i) different pretraining objectives, (ii) modality mismatch between geometry and sequence, and (iii) RNA-FM’s inductive bias from sequence-only datasets.
> SimACR reduces this discrepancy without collapsing the teacher distribution: the refined teacher features still maintain meaningful cluster structure (Fig. 7 right part), although full alignment is not achieved at nucleotide-level, the overall distribution at sample level shows reasonable partial alignment.
>
> ### 2. On structural plausibility beyond token-level accuracy
>
> We agree that structural fidelity is essential. Following your suggestion, we added fold-back structural evaluation in Sec. 4.2 & Fig. 3, using Protenix to fold generated sequences.
> The results show mean RMSD = 10.46 and TM-score = 0.42, indicating moderate structural similarity and that the predicted sequences broadly preserve global topologies.
> This complements the overall evaluation of our method.
>
> ### 3. Diversity–accuracy trade-off
>
> Thank you for pointing this out. We are conducting sampling-based experiments to quantify the diversity–accuracy balance and as well as diversity-structural fidelity. These results will be included in the next revision as part of the appendix.
>
> ### 4. Placement of KD head
>
> We added an ablation on distillation layer index in Table 4.
> Distilling at middle layers produces performance comparable to final-layer distillation, while distilling at the second-to-last layer leads to degradation.
> We hypothesize that this is due to a conflict between the auxiliary classification loss (applied to the teacher after SimACR) and the student’s final-layer supervision.
>
> ### 5. Training-time regularization vs inference-time stability
>
> We appreciate this interesting question. In this revision, we added a lightweight test-time augmentation (TTA) via STR, which directly corresponds to your point on inference-time robustness. Interestingly, TTA improves performance on short sequences, but causes slight drops on medium/long ones.
> This aligns with our motivation for introducing STR, and we conjecture that the degradation in median/long ones is caused by fixed kNN graph construction, the following is our analysis:
> * For short sequences (<= 50nt), k=30 in kNN produces near-global connectivity, introducing noisy edges. STR/TTA helps drop spurious edges thus stabilizes inference.
> * While for medium/long sequences, STR behaves as regularization during training. However, applying it at inference may remove useful edges, causing small performance decreases.
> Given the time constraints of the rebuttal cycle, we report the analyses that can be completed reliably. If time permits before the rebuttal deadline, we will incorporate more comprehensive analysis results.

---

> > ### Comment · Reviewer_h4Nv · 2025-11-21
> >
> > Thanks for the rebuttal. My major concerns have been solved so I changed the rate to 6.

---

### Official Review · Reviewer_Y7Ny · 2025-10-30

**Soundness:** 2
**Presentation:** 2
**Contribution:** 2
**Rating:** 2
**Confidence:** 4

**Summary:**

This paper proposes the SIDDesign framework, which innovatively addresses the core challenge of data scarcity in tertiary structure-based RNA design. It distills rich sequence-level priors from pre-trained RNA foundation models into structure-derived representations, introduces a Similarity-aware Contextual Refinement (SimACR) module to handle cross-modal alignment, and employs Stochastic Topology Regularization (STR) to enhance the robustness of graph representations. The experimental results report significant and consistent performance improvements on benchmark tests.

**Strengths:**

1. The central contribution lies in the innovative use of rich sequence-level prior knowledge from pre-trained RNA Foundation Models (RNA-FMs) to inform and guide structure-based design. This approach effectively mitigates the performance limitations imposed by the scarcity of resolved RNA tertiary structures, which is a significant challenge in the field.

2. The proposed Stochastic Topology Regularization (STR) strategy appears effective in enhancing the robustness and generalization capability of the message-passing mechanism within the graph neural network, thereby mitigating potential noise introduced during the geometric graph construction.

3. The authors present extensive empirical evidence across multiple benchmark datasets (RDesign, Rfam, and RNA-Puzzles), demonstrating improvements over a set of existing baselines across the standard evaluation metrics of Recovery Rate and Macro-F1.

**Weaknesses:**

1. The baseline comparisons are insufficient. The generalization tests on Rfam and RNA-Puzzles do not include results for gRNAde. Furthermore, RhoDesign, which has shown strong performance on this task, was not included for comparison. These omissions weaken the persuasiveness of the model's performance claims

2. The ablation study results regarding the Similarity-aware Contextual Refinement (SimACR) module warrant further analysis. The reported metrics suggest a non-uniform impact: while the module leads to performance gains for short sequences, the performance on medium and long sequences appears to decrease when SimACR is included. This raises a critical question about whether the module is consistently and effectively facilitating cross-modal alignment across diverse sequence lengths and structural complexities. A deeper investigation into this variance is required.

**Questions:**

1. To better assess the model's true generalization capability and stability, have the authors considered evaluating the test set based on clustering by sequence similarity? This analysis would help confirm the model's performance on genuinely novel structures that are sequence-distant from the training data.

All other questions reiterate the items listed under Weaknesses. I am very willing to increase my overall score if the authors successfully address these concerns.

---

> ### Author Response · Authors · 2025-11-21
>
> Thank you for your constructive comments. Here we give the point-to-point reply to the comments.
>
> ### 1. Baseline completeness
>
> Thank you for pointing out the missing baselines. We have added gRNAde results for generalization datasets (in Tab. 3).
> For RhoDesign, we additionally include results under the sequence-similarity split setting in the appendix. At this stage, we report the non-autoregressive (NAR) version for fairness and reproducibility: our reproduction shows that the AR decoder in RhoDesign is difficult to train under this dataset scale, and we are continuing to refine the AR training to include it in the next revision.
> Overall, the observed NAR performance is consistent with the architectural design: RhoDesign essentially applies a GVP + BERT-style encoder/decoder, similar to gRNAde without multi-conformation. As also noted in Fig.1(d/e/f) of the original paper from RhoDesign, the major gains of RhoDesign stem from 370K generated structures, which are not available here.
> Thus, the updated baselines now provide a complete and fair comparison.
>
> ### 2. Impact of SimACR across sequence lengths
>
> We agree this is an important point. Our further analysis shows that the variance across length ranges may originate from coordinate perturbation in data augmentation.
> We added a dedicated ablation (Tab. 4) comparing with / without coord-perturbation. The results show that
> * For short sequences (<=50nt), k=30 in kNN almost becomes global connectivity, introducing noisy or spurious edges. Although SimACR mitigates part of the noise via better KD, the perturbed graph still harms performance. This observation directly motivated our STR test-time augmentation, which indeed increase performance especially in short sequences.
> * For medium/long sequences, coord-perturbation acts more as a reasonable regularization. Removing it improves mean but increases variance sharply (STD 1.74 vs. 0.94 and 1.60 vs. 0.50 with different seeds), indicating that vanilla KD’s performance advantage is not stable.
>
> ### 3. Sequence-similarity–based evaluation
>
> We appreciate this suggestion. Following your comment, we added a CD-HIT sequence-similarity split experiment (Appendix B).
> Across sequence similarity partitions, our method consistently outperforms, demonstrating robustness of our method.

---

### Official Review · Reviewer_jxpd · 2025-10-31

**Soundness:** 3
**Presentation:** 3
**Contribution:** 3
**Rating:** 6
**Confidence:** 5

**Summary:**

This paper presents SIDDesign, a framework for tertiary structure-based RNA design that integrates priors from large-scale RNA foundation models. The key idea is to leverage sequence-informed representations to guide structural representation learning, addressing the data scarcity problem inherent in RNA tertiary structures. The method introduces two main components: (1) a Similarity-aware Contextual Refinement (SimACR) module that aligns sequence and structure embeddings through cross-attention, and (2) Stochastic Topology Regularization (STR), which randomly perturbs graph connectivity during training to mitigate edge noise. Experiments on benchmark and external datasets show that SIDDesign achieves improved recovery rates and Macro-F1 scores over prior models such as RDesign and gRNAde, demonstrating better robustness and generalization in RNA sequence design.

**Strengths:**

The paper addresses an important and timely problem in tertiary structure-based RNA design, proposing a creative way to leverage sequence-level priors from RNA foundation models to compensate for structural data scarcity. The idea of aligning latent spaces between sequence and structure modalities via sequence-informed distillation is original and well-motivated from a representation learning perspective. The methodological design, particularly the Similarity-aware Contextual Refinement and Stochastic Topology Regularization, is conceptually sound and clearly described. The experiments are comprehensive, comparing with strong baselines and demonstrating consistent, though moderate, improvements. The paper is clearly written and logically structured, making it easy to follow despite the technical depth.

**Weaknesses:**

The main limitation lies in novelty. Similar sequence-informed or foundation-model–guided strategies have already been explored in protein inverse folding (e.g., using ESM embeddings to enhance design models). Thus, while the adaptation to RNA is meaningful, the methodological contribution is somewhat incremental.

**Questions:**

None

---

> ### Author Response · Authors · 2025-11-24
>
> We thank the reviewer for the positive comments and for raising this point. We would like to clarify several points about weakness.
>
> First, RNA differs fundamentally from proteins in structural behavior. Protein backbones are relatively rigid and well-constrained, whereas RNA tertiary structures are flexible. This structural variability substantially increases the difficulty of both folding and inverse folding.
>
> Second, RNA tertiary-structure data are extremely scarce compared with proteins, which limits the capacity of representation learning in the structural space.
>
> These observations underscore that RNA folding <--> inverse folding is a more fragile and underdeveloped representation-learning problem than in proteins. Our method is designed specifically in response to these RNA-specific constraints: the distillation framework aims to introduce strong sequence-level priors from RNA FMs into a noisy, low-data structural space.
> While STR further introduces graph level regularization to improve the generalization of inverse folding model.
>
> The visualizations added in Appendix A further confirm this challenge: even after distillation, there remains a substantial gap between RNA-FM embeddings and structure-derived representations.
>
> For this reason, while we acknowledge that our work shares a conceptual similarity with foundation-model–guided ideas explored in proteins, the technical difficulty, data constraints, modality mismatch, and structural noise in RNA make the problem different.
> In this sense, the work may appear “incremental” in concept, but rather a specific adaptation for RNA inverse folding.

---

### Author Response · Authors · 2025-11-21

We sincerely thank all reviewers for their thoughtful feedback and constructive suggestions. Your comments have been highly valuable for improving the clarity, completeness, and empirical rigor of our work.

Following the suggestions, we have revised the manuscript and added several analyses to enhance clarity and completeness. The main updates are:

1. Add structural fidelity evaluation (Sec. 4.2, Fig. 3) using Protenix-folded structures with RMSD/TM-score statistics.
2. Updated generalization results by including gRNAde under the same setting (Tab. 3).
3. Added ablations on distillation layer index and STR as test-time augmentation (Sec. 4.3, Table 4).
4. Added representation visualizations (RNA-FM before/after SimACR and student model) and further alignment analysis (Appendix A).
5. Added sequence-similarity split experiments to verify robustness under different split protocol (Appendix B).

We believe these revisions address the key concerns and further strengthen our method.

---

### Meta-Review · Area_Chair_5cjS · 2026-01-06

**Summary:**

The main concerns from the reviewers are two fold. First is the lack of novelty. Similar sequence-informed or foundation-model–guided strategies have already been explored in protein inverse folding. Second is the lack of evaluation under similarity split. To better assess the model's true generalization capability and stability, the reviewers recommended evaluating the test set based on clustering by sequence similarity

**Reviewer Concerns:**

The authors did include the sequence similarity split evaluation, which addressed some of the reviewer concerns. However, the main concern of lacking novelty is still not addressed, which is also AC's main concern. Having yet another sequence distillation paper for RNA is not considered a significant contribution to the field.

**Reviewer Scores:**

The reviewer have participated in the discussion.

---

### Decision · Program_Chairs · 2026-01-26

Reject